# Development of a New Cell-Based AP-1 Gene Reporter Potency Assay for Anti-Anthrax Toxin Therapeutics

**DOI:** 10.3390/toxins15090528

**Published:** 2023-08-28

**Authors:** Weiming Ouyang, Tao Xie, Hui Fang, David M. Frucht

**Affiliations:** Division of Biotechnology Review and Research II, Office of Biotechnology Products, Office of Pharmaceutical Quality, Center for Drug Evaluation and Research, U.S. Food and Drug Administration, Silver Spring, MD 20993, USA; tao.xie@fda.hhs.gov (T.X.); hui.fang@fda.hhs.gov (H.F.)

**Keywords:** anthrax, *Bacillus anthracis*, lethal toxin, edema toxin, protective antigen, AP-1, reporter assay

## Abstract

Anthrax toxin is a critical virulence factor of *Bacillus anthracis*. The toxin comprises protective antigen (PA) and two enzymatic moieties, edema factor (EF) and lethal factor (LF), forming bipartite lethal toxin (LT) and edema toxin (ET). PA binds cellular surface receptors and is required for intracellular translocation of the enzymatic moieties. For this reason, anti-PA antibodies have been developed as therapeutics for prophylaxis and treatment of human anthrax infection. Assays described publicly for the control of anti-PA antibody potency quantify inhibition of LT-mediated cell death or the ET-induced increase in c-AMP levels. These assays do not fully reflect and/or capture the pathological functions of anthrax toxin in humans. Herein, we report the development of a cell-based gene reporter potency assay for anti-PA antibodies based on the rapid LT-induced degradation of c-Jun protein, a pathogenic effect that occurs in human cells. This new assay was developed by transducing Hepa1c1c7 cells with an AP-1 reporter lentiviral construct and has been qualified for specificity, accuracy, repeatability, intermediate precision, and linearity. This assay not only serves as a bioassay for LT activity, but has applications for characterization and quality control of anti-PA therapeutic antibodies or other products that target the AP-1 signaling pathway.

## 1. Introduction

Anthrax is a life-threatening disease developed following infection with *Bacillus anthracis* [1,2,3,4,5,6]. Anthrax has emerged as a biological terrorism threat [6,7,8], which was highlighted by deadly attacks through the postal service in the form of powdered anthrax spores in 2001. The attacks caused life-threatening infections in 11 patients, 5 of whom succumbed [9,10,11]. Disease pathology is mediated by the pXO1 virulence plasmid of *B. anthracis*, which expresses the anthrax toxin components, including protective antigen (PA) and two enzymatic moieties, edema factor (EF) and lethal factor (LF) [1,12,13]. The PA toxin component is an 83-kDa protein with four functional domains [14,15]. Following binding via its fourth domain to its cellular receptors, TEM8 and CMG2, the first domain of PA is cleaved by cell surface protease such as furin, leading to oligomerization of the remaining PA (PA63) to form the LF/EF binding sites. The receptor–toxin complex is then internalized by clathrin-dependent endocytosis, and in the low pH environment, the conformation of PA changes from a prepore structure to an SDS- and heat-stable pore structure, which allows LF and EF translocation to the cytosol. In the cytosol, LF acts as a zinc-dependent metalloprotease that specifically cleaves mitogen-activated protein kinase kinases (MKKs) [16,17,18], PI-3K regulatory subunit p110 [19] and, in certain mouse and rat strains, NACHT leucine-rich repeat protein 1 (NLRP1) [20,21,22,23], a protein involved in cell death pathways. EF functions as a Ca^2+^ and calmodulin dependent adenylate cyclase that increases the level of cAMP in the host cell, leading to disruption of water homeostasis, edema and tissue damage [12,24]. 

Anthrax toxin causes many of the systemic injuries associated with anthrax pathophysiology [3,9,11,12]. It has been reported that death ensues even if sterility of the bloodstream is achieved with antibiotics once toxin levels have reached a critical threshold in experimental animals [25]. Due to the critical role of the PA in mediating the toxin entry, neutralizing antibodies targeting PA have been developed for clinical use [26]. One example is raxibacumab, a human anti-PA monoclonal antibody that specifically recognizes the fourth domain of the PA protein with a high affinity and blocks the binding of PA to anthrax toxin receptors [27]. Raxibacumab was approved by the FDA in December 2012 for treatment of inhalational anthrax due to *Bacillus anthracis* in combination with appropriate antibacterial drugs, and for prophylaxis of inhalational anthrax when alternative therapies are not available or are not appropriate [15,28,29,30,31,32]. In March 2016, the FDA approved a second anti-PA antibody, obiltoxaximab (Anthim), for treatment of inhalational anthrax in combination with appropriate antibacterial drugs [14,33,34,35].

As with all monoclonal antibodies approved for use in humans, the control of product quality is essential. To this end, the use of a cell-based potency assay that reflects the presumed mechanism of action (MoA) of the product in vivo is expected. The potency assays for assessing anthrax toxin neutralization activity of anti-PA antibodies described in the scientific literature include methods that are based on either LT-induced cell death via NLRP1 or the EF-induced increase in the level of c-AMP [36,37,38,39]. However, these assays may not fully reflect and/or capture the pathological functions of anthrax toxin in humans because LT does not cleave human NLRP1, and ET may play a less important role than LT in the pathogenesis following infection with *Bacillus anthracis* [40,41,42]. Therefore, development of a new cell-based bioassay that fully reflects the pathological function of anthrax toxin in humans will facilitate the control of potency of anti-PA antibodies.

Studies from our laboratory have revealed that anthrax LT has a potent degradatory effect on c-Jun oncoprotein, which is a major member of the activation protein 1 (AP-1) transcription factor family. c-Jun regulates numerous cellular activities in mammals [43,44,45]. Inactivation of the MKK1/2–Erk1/2 signaling pathway following LT treatment induces a rapid reduction in levels of the c-Jun protein by promoting its degradation via a COP1-dependent pathway [18,46,47]. In this study, we developed a cell-based AP-1 reporter assay for the potency control of anti-PA antibodies that is based on the rapid reduction of c-Jun protein levels in LT-treated cells, an effect relevant to human cells. This new AP-1 reporter potency assay has been qualified for specificity, accuracy, repeatability, intermediate precision, and linearity. 

## 2. Results

### 2.1. Optimization of LT and Anti-PA Antibody Concentrations for Hepa1c1c7 Cell Treatment 

Potency assays described in the literature for anti-anthrax toxin therapeutic antibodies do not fully reflect the pathological functions of anthrax toxin in humans. To develop a cell-based AP-1 reporter assay for anti-anthrax toxin therapeutic antibodies, we first generated a Hepa1c1c7 cell line that stably expresses green fluorescent protein (GFP) and an AP-1 gene reporter by transducing Hepa1c1c7 cells with the pseudo lentiviral pGreenFire 2.0 AP-1 reporter construct. Transduced cells were cultured in selection medium containing 10 µg/mL puromycin, and the resulting bulk stable cell line expressing GFP and the AP-1 gene reporter was designated as Hepa1c1c7-AP1Luc. To optimize the concentration of anthrax LT for the assay, we first treated the Hepa1c1c7-AP1Luc cells with a fixed concentration of PA (1000 ng/mL) in combination with two-fold serially diluted concentrations of LF from 1000 ng/mL to 1.95 ng/mL. Treated cells were extracted and assessed for luciferase activity following LT treatment for 24 h. As shown in Figure 1A, the lowest concentration of LF required to reach the maximum inhibition of AP-1 activity was 7.8 ng/mL. To ensure a maximum level of AP-1 inhibition, we chose 20 ng/mL as a fixed concentration for LF for the assay. To evaluate the appropriate concentration of PA for the assay, Hepa1c1c7-AP1Luc cells were then treated with 20 ng/mL LF in combination with two-fold serially diluted concentrations of PA from 160 ng/mL to 20 ng/mL. All treatments caused a maximum inhibition of AP-1 activity. As the molecular weight of PA is approximately twice that of LF, 20 ng/mL LF and 40 ng/mL PA were chosen for further development of the AP-1 gene reporter assay to reflect equimolar concentrations. We next treated the cells with LT in combination with serially two-fold diluted anti-PA antibody to assess the protective activity of the antibody. As shown in Figure 1C, treatment with anti-PA antibody resulted in a dose-dependent protection of AP-1 levels in Hepa1c1c7 cells treated with LT. 

### 2.2. Development of Single Cell Clones of Hepa1c1c7-AP1Luc Cells for the AP-1 Reporter Assay

We considered the possibility that Hepa1c1c7-AP1Luc bulk cells could express variable levels of luciferase and respond differently to LT and anti-PA antibody treatment, thereby leading to high assay variation. To assure consistency of the responsiveness of Hepa1c1c7-AP1Luc to LT and anti-PA antibody treatment and reduce assay variation, we generated single cell clones from Hepa1c1c7-AP1Luc parental cells using FACS-sorting and the limiting dilution cloning method [48]. Six clones harboring various levels of basal luciferase activity were selected to test their responsiveness to LT treatment. As shown in Figure 2A, the AP-1 activities in all six clones were inhibited in a dose-dependent manner following 24-h LT treatment. The maximum inhibition was reached in all six clones when treated with 16 ng/mL LF and PA for 24 h, which was in concordance with the concentration of LT optimized using the bulk cells. Interestingly, Clone 3D5 maintained a relatively high level of AP-1 activity following LT treatment (67.20% inhibition for 3D5 vs. 87.98 ± 2.32 average inhibition for the other five clones). Clones 3C12 and 1H3 had the same levels of basal luciferase activity and responded similarly to LT treatment. For further testing of the response to anti-PA antibody, we excluded 3D5 and 3C12, and treated the other four clones with 20 ng/mL LF and 40 ng/mL PA together with 2-fold serially diluted anti-PA antibody. Each of these four clones showed a dose-dependent sigmoidal curve response to anti-PA antibody (Figure 2B–E). The lowest concentration of anti-PA antibody required to reach the maximum protection of AP-1 activity in LT-treated cells was 20–40 µg/mL (Figure 2B–E). Based on its basal luciferase activity and responsiveness to LT and anti-PA antibody treatment, clone 1H3 was selected for further development of the AP-1 reporter potency assay.

### 2.3. Optimization of the Cell Density for the AP-1 Reporter Assay

We reasoned that the number of cells that are initially introduced into each well of a 96-well plate may subsequently impact their future cell growth, basal level of luciferase activity, and responses to LT and anti-PA antibody treatment. To optimize the cell number for the assay, we placed varying numbers of 1H3 cells (10, 20, or 30 thousand) into each well of 96-well plates. The cells were then treated with LT together with serially two-fold diluted anti-PA antibody. The 1H3 cells cultured with the three different initial densities exhibited similar sigmoidal curve responses in anti-PA-mediated protection of AP-1 activities in LT-treated cells with different upper asymptotes (Figure 3A–C). Based on the upper asymptote and the microscopic observation that wells with 30 thousand cells became confluent following culture for one day, an initial cell count of 20 thousand cells per well was chosen for further qualification of the AP-1 reporter potency assay.

### 2.4. Qualification of the AP-1 Reporter Potency Assay

Following titration of LT and anti-PA antibody and optimization of the initial cell density, we next performed qualification studies for the AP-1 reporter potency assay. The pattern of anti-PA-mediated protection of AP-1 activity in LT-treated 1H3 cells is a sigmoidal standard curve. This pattern allowed us to measure the potency of anti-PA antibody relative to a reference standard using the four-parameter logistic (4PL) sigmoidal model and comparing the EC_50_ values of the tested samples and the reference standard. Based on the dose response of 1H3 cells shown in Figure 3, we used 12 serially diluted (2-fold) concentrations of anti-PA antibody samples and the reference standard to qualify the AP-1 reporter potency assay for accuracy, repeatability, intermediate precision, and linear range. The AP-1 reporter potency assay had −0.93%, 1.31%, −3.64%, 3.22%, and −3.42% relative bias (RB), and 5.95%, 7.41%, 4.70%, 4.10%, and 6.42% coefficient of variation (CV) at the relative potency (RP) levels of 50%, 75%, 100%, 125%, and 150%, respectively (Figure 4A and Table 1). The 0.9981 R^2^ of the linear regression supported a linear range of the assay from 50% RP to 150% RP (Figure 4B and Table 2). The intermediate precision of the AP-1 reporter potency assay was established by 14 datasets generated from 7 independent assay runs, each performed by 2 analysts independently. The intermediate precision of the assay at the 100% relative potency level was 6.79% (Table 2).

We also qualified the assay for specificity. 1H3 cells were treated with LT together with anti-PA antibody or normal control human IgG. The anti-PA antibody but not the control human IgG protected AP-1 activity in LT-treated cells (Figure 4C and Table 2), and 4 logistic parameters converged when the assay was performed using anti-PA antibody but not the normal human IgG. These results support the conclusion that the AP-1 reporter assay specifically detects the activity of anti-PA antibody to protect AP-1 levels in LT-treated cells.

### 2.5. Stability-Indicating Capacity of the AP-1 Reporter Potency Assay

Following qualification of the AP-1 reporter potency assay, we next investigated whether the assay could detect a change in the potency of stressed anti-PA antibody samples. To this end, anti-PA antibody was incubated at 40 °C, 50 °C, or 60 °C for 72 h. The stressed samples were then assessed for potency together with a reference standard and an unstressed sample using the AP-1 reporter potency assay. As shown in Figure 5, the stressed sample incubated at 60 °C for 72 h lost most of its activity, exhibiting only 4.19 ± 0.80% RP. These results demonstrate that the AP-1 reporter potency assay can detect a change in the potency of stressed anti-PA samples and may serve as a stability-indicating assay for anti-PA therapeutic antibodies.

## 3. Discussion

Anti-anthrax toxin antibodies are approved by the FDA for prophylaxis and treatment of inhalational anthrax due to *Bacillus anthracis* infection [14,15,28,29,30,31,32,33,34,35]. Herein, we reported the development of a new cell-based AP-1 reporter potency assay for anti-anthrax toxin therapeutic antibodies, based on their neutralizing activity on a specific effect of anthrax toxin conserved among mammalian cell targets. Anti-anthrax toxin antibodies target PA [14,15], the toxin component mediating entry of the enzymatic LF and EF into the cytosol where they execute pathological functions as a metalloprotease [16,17,18] and an adenylate cyclase [12,24], respectively. 

Appropriate control of the critical quality attributes (CQA) is essential to ensure the consistent quality of therapeutic biologics such as monoclonal antibodies targeting anthrax toxin. Potency is one such CQA. Assays described in the publicly available literature for anti-PA antibodies include a cAMP assay that was developed based on ET’s calmodulin-dependent adenylyl cyclase activity to elevate intracellular cAMP levels by converting ATP to cAMP, and a toxin neutralization assay (TNA) that was developed on the basis of a strain-specific effect on macrophage cell lines induced by LT-activation of NLRP1 inflammasome [36,37,38,39]. 

Although administration of both LT and ET can cause animal death, ET may have a less important role than LT in the lethality caused by anthrax infection. LF and EF are found to exist in vivo at a ratio of 5:1 in anthrax-infected animal models [40,41], and a lethal dose of ET may only be reached at a very late stage of anthrax infection. Using the GI anthrax mouse model [42,49], we also observed that anthrax LT is required to establish lethal infection, whereas anthrax ET modulates progression and dissemination of infection but not lethality, corroborating the notion that LT has a more important role than ET in anthrax pathogenesis. These findings suggest that the cAMP potency assay may not sufficiently reflect the in vivo MoA of anti-PA therapeutic antibodies. 

The TNA assay was developed based on mouse macrophage cell lines RAW264.7 and J774, which are uniquely sensitive to LT and are killed by LT in less than two hours. The species- and strain-specific sensitivity of rodent macrophages to LT is due to polymorphisms in rat *Nlrp1* and mouse *Nlrp1b* [20,21,22,23]. *Nlrp1* and *Nlrp1b* encode the NOD-like receptor (NLR) protein NLRP1, which controls activation of caspase-1 by functioning as the sensor component of the multiprotein NLRP1 inflammasome. LF activates the NLRP1 inflammasomes in LT-sensitive rat and mouse strains via direct cleavage of the NLRP1 and NLRP1b proteins at sites near their amino-termini [20,21], leading to a caspase 1-mediated rapid macrophage cell death. Although primary macrophages from certain mouse and rat strains (such as BALB/c mice and Fischer rats) also exhibit high sensitivity to LT, macrophages from other mouse and rat strains (such as C57BL6 mice and Lewis rats), as well as other mammals are resistant. Therefore, the TNA potency assay does not reflect the in vivo MoA of anti-PA therapeutic antibodies in humans.

Our new AP-1 reporter assay is based on the signal pathway immediately downstream of initial MKK targets of anthrax LT, an intracellular target shared across species, including humans. The MKKs are the middle layer of three-tiered signaling cascades triggered by extracellular stimuli such as growth factors or cytokines. Extracellular stimuli initiate activation of mitogen-activated protein kinase kinase kinases (MKKKs) that phosphorylate MKKs, which, in turn, phosphorylate mitogen-activated protein kinases (MAPKs). LF blocks the activation of Erk1/2, p38, and JNK MAPKs by direct cleavage of MKKs at their docking sites (D sites) [16,17,18], disrupting numerous biological responses in mammalian cells such as cell growth, cell development, cell differentiation, cell survival, and inflammatory immune responses [50]. Our cell-based potency assay has been developed based on the AP-1 activity in LT-treated cells, which is regulated by both Erk1/2 and JNKs. LT also cleaves MKK3, the kinase activating p38, with similar efficiency as compared to its cleavage of MKK1, MKK2, and MKK4 that activate Erk1/2 and JNKs. Therefore, this cell-based potency assay may also indirectly reflect the disruption of p38 signaling. Given the essential role of the MAPK signaling pathways in mammalian cells, we believe that our new potency assay may better reflect the in vivo MoA than assays described in the literature for the control of anti-PA antibody potency. 

Moreover, we have qualified our assay for accuracy, repeatability, linearity, intermediate precision, and specificity. Assay qualification results show that this new AP-1 reporter assay has similar repeatability and intermediate precision when compared to the published cAMP assay [37] and TNA [38]. However, the cAMP assay was validated using a self-developed human-anti-PA monoclonal antibody [37] and the TNA was validated using human anti-anthrax vaccine adsorbed (AVA) [38]; therefore, a side-by-side study using the same testing material would be necessary to definitively compare AP-1 reporter, TNA, and cAMP assay performance.

Additionally, we have shown that the AP-1 reporter assay has the capability to detect potency changes in stressed anti-PA samples. Furthermore, during qualification of the assay, we optimized the initial cell concentration and utilized precision methods to minimize assay variability. Together, the consistent responsiveness and method controls provide assurance of a high performance of the AP-1 reporter assay for the control potency. The new high-performing AP-1 reporter assay may be used for evaluation of the potency of a variety of therapeutics targeting PA, including other licensed mAbs (e.g., obiltoxaximab) or Anthrax Immunoglobulin Intravenous (AIGIV) [26,51]. Lastly, this new AP-1 reporter assay may also warrant investigation for use in evaluation of the immunogenicity of anthrax vaccines by measuring the anti-PA titers in humans following vaccination of BioThrax AVA and other recombinant PA vaccines [52], once validated for use in a serum matrix. 

## 4. Materials and Methods

### 4.1. Cell Line and Reagents for Development of the AP-1 Reporter Assay

Murine hepatoma Hepa1c1c7 cells (CRL-2026™) purchased from the American Type Culture Collection (ATCC, Manassas, VA, USA) were cultured in an alpha minimum essential medium (MEM), supplemented with 10% FBS, 100 IU/mL penicillin, 100 µg/mL streptomycin, and 2 mM L-glutamine. Cultures were maintained at 37 °C in an incubator with 5% CO_2_. Lyophilized recombinant protective antigen (PA) and lethal factor (LF) were purchased from List Biological Laboratories, Inc. (Campbell, CA, USA), and reconstituted in sterile water to make stock solutions with a final concentration of 1 mg/mL for LF and 2 mg/mL for PA. The stock solutions were then diluted 10 times with 50% glycerol in sterile water to make stock solutions with a final concentration of 0.1 mg/mL for LF and 0.2 mg/mL for PA. The solutions were aliquoted to 25 µL in each 0.5 mL Eppendorf protein LoBind tube and frozen stored in a −80 °C freezer. The anti-PA monoclonal antibody used for qualification studies was raxibacumab (GlaxoSmithKline, Brentford, UK), which had been cycled off from the Strategic National Stockpile after surpassing its expiration period. Biological activity was confirmed to be retained in the expired anti-PA antibody using an in vitro IL-2 production inhibition assay [53,54]. An aliquot of anti-PA antibody was taken from the glass vial and diluted with PBS to make a batch of reference standard with a concentration of 40 mg/mL, which was aliquoted into 0.5 mL Eppendorf protein LoBind tubes (100 µL/tube) and stored at −80 °C. 

### 4.2. Transduction and Establishment of the Hepa1c1c7-AP1Luc Cell Line for the AP-1 Reporter Assay 

Hepa1c1c7 cells were transduced with pGreenFire 2.0 AP-1 reporter virus (pGF2-AP1-rFluc-T2A-GFP-mPGK-Puro) purchased from System Biosciences (Palo Alto, CA, USA) following the instructions provided the vendor. Transduced cells were cultured in alpha MEM selection medium, which was supplemented with 10% FBS, 100 IU/mL penicillin, 100 g/mL streptomycin, 2 mM L-glutamine, and 10 µg/mL puromycin for 2 weeks and then FACS-sorted for GFP-positive cells. Following culture in the selection medium for one week, the sorted cells were FACS-sorted for a second time and then maintained in the selection medium. The sorted GFP-positive bulk cells were designated as Hepa1c1c7-AP1Luc. 

### 4.3. Development of Single Cell Clones and a Cell Bank for the AP1 Reporter Assay

Hepa1c1c7-AP1Luc cells were used to generate single cell clones by the limiting dilution cloning method [48]. The cell suspensions were diluted in the selection medium to reach a final concentration of 5 cells/mL and 100 µL of the diluted cells was added to each well of 96-well plates. Cell growth was monitored, and single cell clones were chosen by microscopy. Cells derived from each single clone were transferred to two separate wells of a 24-well plate. Cells in one well were used for analysis of luciferase activity and cells in the other well were used for expansion. Following screening for luciferase activity and cell growth, 6 single clones were selected for further testing of their responses to LT treatment and then 4 clones were tested for their responses to anti-PA antibody treatment. Clone 1H3 was chosen as the clone used for subsequent development following screening tests. Clone 1H3 cells were expanded in the selection medium to generate an 80-vial cell bank, which was stored in a liquid nitrogen tank. 

### 4.4. Cell Treatment and Measurement of Luciferase Activity in Treated Cells

Hepa1c1c7-AP1Luc cells and single cell clones were treated with or without LT in combination with or without anti-PA antibody. Cells were then lysed with the Passive lysis buffer for luciferase activity measurement using the Promega Luciferase Assay System (Cata# E4550) and the Glomax 96 microplate luminometer purchased from Promega (Madison, WI, USA). Triplicate wells were used for each treatment. AP-1 activity inhibition of LT-treated cells was calculated using the following formula: (luminescence of untreated cells—luminescence of treated cells) ÷ luminescence of untreated cells × 100%. The anti-PA antibody protection of AP-1 activity in LT-treated cells was calculated using the following formula: (luminescence of cells treated with LT and anti-PA antibody—luminescence of LT-treated cells) ÷ (luminescence of untreated cells—luminescence of LT-treated cells) × 100%. For potency measurements, the ONE-Glo^TM^ Luciferase Assay System (Cata# E6120) and the Glomax Navigator microplate luminometer were used to measure the luciferase activity in the treated 1H3 cells. Measurements were performed following the manufacturer’s instructions. 

### 4.5. Measurement of Anti-PA Potency

To measure the relative potency of anti-PA antibody samples, 1H3 cells were treated with 20 ng/mL LF and 40 ng/mL PA in combination with 12 serially 2-fold diluted concentrations of anti-PA reference standard or test samples from 160 µg/mL to 80 ng/mL. Each run was performed using triplicate 96-well plates, and electronic pipettes were utilized to maintain maximal precision. Following 24 h of treatment, the cells were extracted for measurements of luciferase activity. The luminescence data were analyzed by Prism GrapPad Prism software version# 9.3.1 (Dotmatics, Boston, MA, USA), and the relative potency was presented as a mean of three data sets from the triplicate plates calculated using the following formula: EC_50_ of the reference standard ÷ EC_50_ of the test sample × 100%. A run was considered invalid if the data failed to pass the parallelism test or the triplicate data had a coefficient of variation (CV) over 35%. 

### 4.6. Qualification of the AP-1 Reporter Assay

The potency of the anti-PA antibody was measured as described above at the relative potency levels of 50%, 75%, 100%, 125%, and 150% seven times. The data from the seven independent runs were analyzed by Prism GrapPad software (Dotmatics) and Microsoft Excel software for relative bias (RB) and CV to demonstrate the assay accuracy and repeatability, and for R^2^ to demonstrate the linearity of the assay. For assay intermediate precision, a different analyst performed the potency measurement of anti-PA antibody at the 100% relative potency level. A total 14 of datasets from 7 runs performed by each of the 2 analysts were used to calculate the CV.

## Figures and Tables

**Figure 1 toxins-15-00528-f001:**
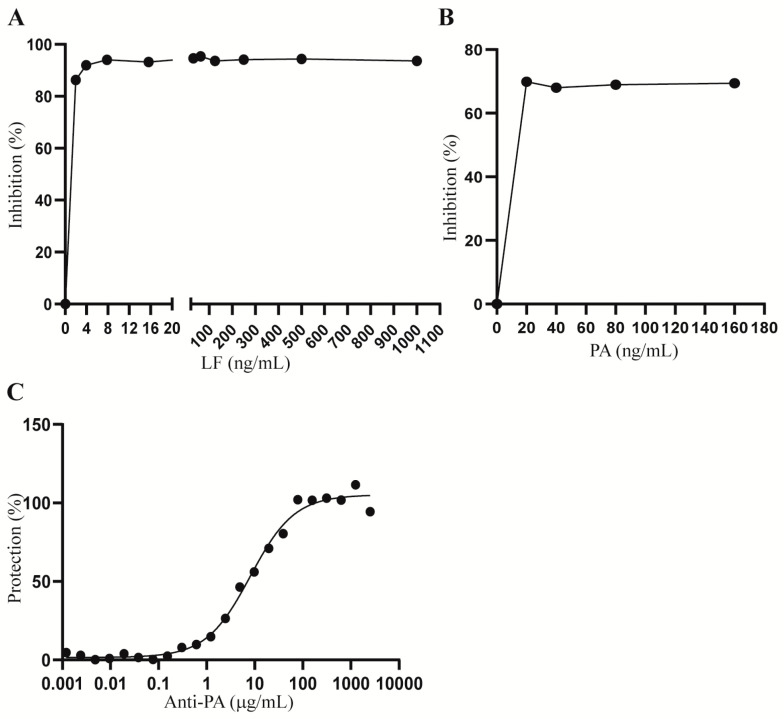
Optimization of lethal toxin and anti-PA concentrations for the treatment of Hepa1c1c7 cells. (**A**) Hepa1c1c7-AP1Luc cells were treated with 1000 ng/mL PA in combination with serially two-fold diluted concentrations of LF ranging from 1000 ng/mL to 1.95 ng/mL. (**B**) Hepa1c1c7-AP1Luc cells were treated with 20 ng/mL LF in combination with serially two-fold diluted concentrations of PA ranging from 160 ng/mL to 20 ng/mL. (**C**) Hepa1c1c7-AP1Luc cells were treated with 20 ng/mL LF and 40 ng/mL PA in combination with 22 serially two-fold diluted concentrations of anti-PA ranging from 5000 µg/mL to 1 ng/mL. Following 24 h of treatment, the cells were extracted for measurement of luciferase activity using the Luciferase Assay System and Glomax 96 microplate luminometer from Promega (Madison, WI, USA). Results are presented as inhibition % (**A**,**B**) or protection % (**C**). Data shown are representative of two independent experiments.

**Figure 2 toxins-15-00528-f002:**
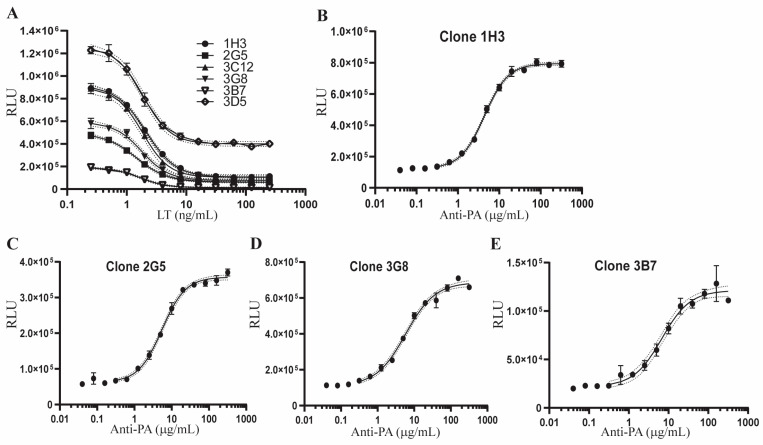
Responsiveness of Hepa1c1c7-AP1Luc single cell clones to LT treatment and anti-PA antibody protection. (**A**) Six single cell clones were generated from sorted GFP-positive Hepa1c1c7-AP1Luc cells by the limiting dilution cloning method. The six clones were treated with 11 serially 2-fold diluted concentrations of LT ranging from 250 ng/mL to 0.25 ng/mL. (**B**–**E**) Single cell clones of 1H3, 2G5, 3G8, and 3B7 were treated with 20 ng/mL LF and 40 ng/mL PA in combination with 14 serially 2-fold diluted concentrations of anti-PA ranging from 320 µg/mL to 40 ng/mL. Following 24 h of treatment, the cells were extracted and assessed for luciferase activity using the Luciferase Assay System and Glomax 96 microplate luminometer from Promega. Results are presented as relative luminescence units (RLU). Error of the interpolated curve is represented by the dot lines adjacent to the solid line. Data shown are representative of two independent experiments.

**Figure 3 toxins-15-00528-f003:**
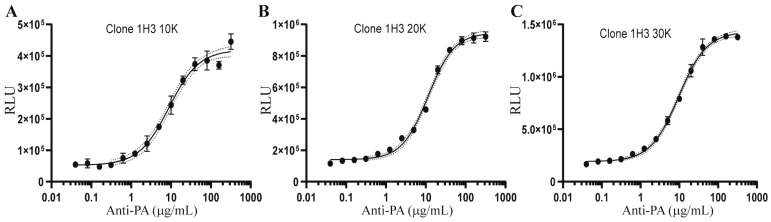
Optimization of the cell density for the AP-1 reporter assay. Totals of 10 (**A**), 20 (**B**), or 30 (**C**) thousand clone 1H3 cells were added to each well of 96-well plates. The cells were treated with 20 ng/mL LF and 40 ng/mL PA in combination with 14 serially 2-fold diluted concentrations of anti-PA ranging from 320 µg/mL to 40 ng/mL. Following 24 h of treatment, the cells were extracted for measurements of luciferase activity using the Luciferase Assay System and Glomax 96 microplate luminometer from Promega. Results are presented as relative luminescence units (RLU). Error of the interpolated curve is represented by the dot lines adjacent to the solid line. Data shown are representative of two independent experiments.

**Figure 4 toxins-15-00528-f004:**
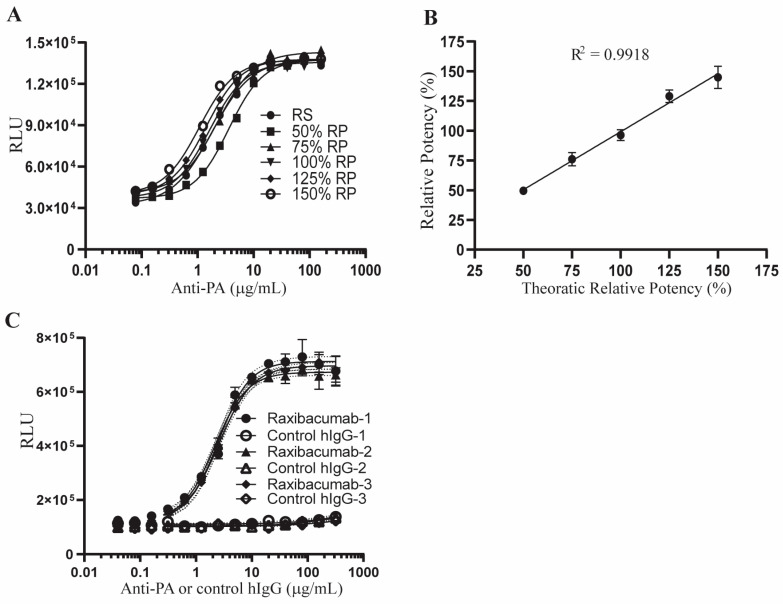
Linear range and specificity of the AP-1 reporter potency assay. (**A**,**B**) The potency of anti-PA antibody was tested using reference standard samples with theoretical 50%, 75%, 100%, 125%, and 150% relative potencies using the AP-1 reporter assay. Data shown in (**A**) are representative of 7 individual experiments and data shown in (**B**) represent the mean ± SEM of the results from the 7 independent experiments. (**C**) A total of 20,000 1H3 cells were treated with 20 ng/mL LF and 40 ng/mL PA in combination with 14 serially 2-fold diluted concentrations of anti-PA antibody or control human IgG ranging from 320 µg/mL to 40 ng/mL. Following 24 h of treatment, the cells were extracted for luciferase activity measurements using the Luciferase Assay System and Glomax 96 microplate luminometer from Promega. Results are presented as relative luminescence unit (RLU). Error of the interpolated curve is represented by the dot lines adjacent to the solid line. Data shown are results from three independent experiments as labeled 1, 2, 3 in the figure.

**Figure 5 toxins-15-00528-f005:**
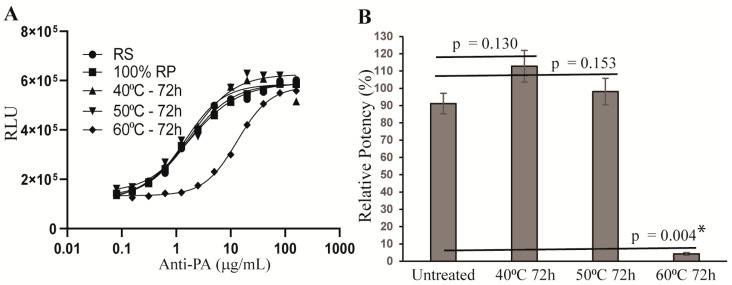
Potency of stressed anti-PA antibody samples measured using the AP-1 reporter assay. Anti-PA antibody was incubated at 40 °C, 50 °C, or 60 °C for 72 h. The stressed samples were assessed for potency together with a reference standard and an unstressed sample using the AP-1 reporter assay. Data shown in (**A**) are representative of three independent experiments, and data shown in (**B**) are the means ± SEM of the results from the three independent experiments. * depicts a statistically significant difference (*p* ≤ 0.05) between the potencies of untreated sample and the sample incubated at 60 °C for 72 h.

**Table 1 toxins-15-00528-t001:** Repeatability and intermediate precision of the cell-based AP-1 gene reporter assay.

**Repeatability**	**Test Results**										
**Theoretical RP**	**Exp. 1**	**Exp. 2**	**Exp. 3**	**Exp. 4**	**Exp. 5**	**Exp. 6**	**Exp. 7**	**Average**	**SD**	**RB%**	**CV**
50	46.68	51.42	45.33	49.27	49.83	49.94	54.27	49.53	2.95	−0.93	5.95
75	71.39	86.74	74.05	79.53	73.44	70.47	76.26	75.98	5.63	1.31	7.41
100	95.49	93.57	101.21	102.26	97.41	95.58	88.98	96.36	4.53	−3.64	4.70
125	123.09	125.32	129.06	126.69	126.74	138.00	134.28	129.03	5.29	3.22	4.10
150	152.88	149.39	155.5	130.85	136.83	149.77	138.91	144.88	9.29	−3.42	6.42
**Intermediate Precision**		**Test Results**									
**Theoretical RP**	**Analyst**	**Exp. 1**	**Exp. 2**	**Exp. 3**	**Exp. 4**	**Exp. 5**	**Exp. 6**	**Exp. 7**	**Average**	**SD**	**CV**
100	1	95.49	93.57	101.21	102.26	97.41	95.58	88.98			
100	2	99.83	103.02	116.27	108.98	94.97	101.30	101.83	100.05	6.80	6.79

**Table 2 toxins-15-00528-t002:** Summary of qualification results for the AP-1 potency assay.

Attribute	Experiment	Acceptance Criteria	Results	Pass/Fail
Accuracy	Reference standard (RS) assessed at five relative potency (RP) levels (50%, 75%, 100%, 125% and 150% by 7 independent experiments	±25% relative bias (RB)	−3.64 ≤ RB ≤ 3.22	Pass
Repeatability		CV ≤ 25%	CV ≤ 7.41%	Pass
Linearity		R² ≥ 0.97	R² = 0.9918	Pass
Intermediate precision	RS assessed at the 100% RP level by 2 analyst and 7 independent experiments for each analyst	CV ≤ 30%	CV = 6.79	Pass
Specificity	Anti-PA antibody and normal human IgG were tested at same concentrations by 3 independent experiments	4-parameter logistic (4PL) convergence for Anti-PA antibody but not control human IgG	4PL convergence for anti-PA antibody, but not for control human IgG	Pass

## Data Availability

The data presented in this study are available in this article as main Figures or Tables.

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
