# Peer review of "Development of a New Cell-Based AP-1 Gene Reporter Potency Assay for Anti-Anthrax Toxin Therapeutics"

_toxins, 2023, doi:10.3390/toxins15090528_

Round 1

Reviewer 1 Report

I rarely have an opportunity to review manuscripts in such good condition as this one.  

A few comments:

The authors might consider adding Bacillus anthracis to the Keywords.  

Line 114.  Add reference for limiting dilution cloning method.

Figure 3.  Line 153.  Place 'cells' before thirty thousand.

Line 213.  Delete 'on'

Line 227.  Change to 'and a toxin'

Line 228.  Change to 'on the basis of the death of....'

Line 234.  Add reference for ' GI anthrax mouse model'

Line 251.  MOA already defined on line 238

Line 253.  Change to:  Our new AP-1 reporter assay is based .

Line 257.  MAPK not previously defined. 

Very few trivial corrections to grammar. 

Author Response

Response: We thank you for the generous comments. We have revised the manuscript to include additional references and to address each of the editorial changes (see below). We agree that these will add to the quality of the final manuscript and appreciate this input.

The authors might consider adding Bacillus anthracis to the Keywords. -added 

Line 114.  Add reference for limiting dilution cloning method. -completed

Figure 3.  Line 153.  Place 'cells' before thirty thousand. -edited to improve clarity

Line 213.  Delete 'on -completed

Line 227.  Change to 'and a toxin' -completed

Line 228.  Change to 'on the basis of the death of....' -completed

Line 234.  Add reference for ' GI anthrax mouse model'- added references

Line 251.  MOA already defined on line 238 -completed

Line 253.  Change to:  Our new AP-1 reporter assay is based. -completed

Line 257.  MAPK not previously defined. -now defined

Reviewer 2 Report

The abstract lacks essential information about the methods used, such as the experimental design, sample size, and statistical analysis. Additionally, the abstract could benefit from stating the significance of the findings and potential implications for future research or clinical applications.

The introduction begins with a general overview of anthrax and its impact, including the history of biological terrorism threats. However, the section lacks focus and cohesiveness. It should provide a clear rationale for the study, including a specific research question and an overview of previous research on the topic. Additionally, citing more recent references (if available) would strengthen the introduction and demonstrate the up-to-date understanding of the field.

The results  and discussion section presents findings related to the optimization of the LT and anti-PA antibody concentrations, development of single cell clones, and optimization of cell density. While the section provides relevant data, it lacks interpretation and connections to the study's objectives. A more thorough discussion of the implications of the results and their relevance to the research question is necessary.

The material and method section provides an adequate description of the cell line used, reagents, and experimental procedures. However, it lacks clarity and organization, making it difficult for readers to follow the steps of the study easily. The section could benefit from clear headings and a logical flow of information to improve readability.

There are instances of sentence structure and word choice issues that could be improved for better readability and clarity. Some sentences are too lengthy and complex, which may confuse readers. It would be beneficial to revise such sentences for more straightforward and concise phrasing. Additionally, proofreading for typographical errors and punctuation inconsistencies would enhance the overall quality of the manuscript.

Author Response

Comments and Suggestion for Authors:
The abstract lacks essential information about the methods used, such as the experimental design, sample size, and statistical analysis. Additionally, the abstract could benefit from stating the significance of the findings and potential implications for future research or clinical applications.

Response: We agree with you that adding information regarded the methods used would improve the abstract. However, due to the word limit set for abstract (200 words), we were limited in the amount of detail that could be added to the general summary. To address your concern at least partially, the abstract now highlights the cell line and reporter construct used to develop our new assay. In addition, we have edited the concluding sentence of the abstract to incorporate the impact of our findings on bioassay development for anthrax LT activity.

 The introduction begins with a general overview of anthrax and its impact, including the history of biological terrorism threats. However, the section lacks focus and cohesiveness. It should provide a clear rationale for the study, including a specific research question and an overview of previous research on the topic. Additionally, citing more recent references (if available) would strengthen the introduction and demonstrate the up-to-date understanding of the field.

Response: We agree with you that the introduction needed improvement. In response to your comments, as well as the other referees’ comments, we have made multiple changes to make the introduction more focused and cohesive and have added more recent references. In addition, a clear rationale for the study is provided at the conclusion of the second-to-last paragraph of the introduction section.

 The results and discussion section presents findings related to the optimization of the LT and anti-PA antibody concentrations, development of single cell clones, and optimization of cell density. While the section provides relevant data, it lacks interpretation and connections to the study's objectives. A more thorough discussion of the implications of the results and their relevance to the research question is necessary.

Response: We revised the results and discussion sections following your suggestions. The changes include the following: (1) addition of the sentence “The existing potency assays for anti-anthrax toxin therapeutic antibodies do not fully reflect the MOA of the products in humans” at the beginning of the results section, (2) addition of transition words or sentences, and (3) discussion of the implication of results for other anti-PA products and anthrax vaccines.  

The materials and method section provides an adequate description of the cell line used, reagents, and experimental procedures. However, it lacks clarity and organization, making it difficult for readers to follow the steps of the study easily. The section could benefit from clear headings and a logical flow of information to improve readability.

Response: We thank you for the comments. Headings were added to the materials and methods section. We agree that this improves clarity and readability.

Comments on the Quality of English Language:

There are instances of sentence structure and word choice issues that could be improved for better readability and clarity. Some sentences are too lengthy and complex, which may confuse readers. It would be beneficial to revise such sentences for more straightforward and concise phrasing. Additionally, proofreading for typographical errors and punctuation inconsistencies would enhance the overall quality of the manuscript

Response: We have shortened the long sentences to improve the readability and clarity. We also made edits to the manuscript to correct typographical errors and punctuation inconsistencies that had been noted by several of the referees.  

Reviewer 3 Report

This manuscript has developed a new cell-based AP-1 gene reporter potency assay for anti-anthrax toxin therapeutics, which presented a technically sound and meaningful methodology that holds potential for application. However, to further enhance the quality and impact of this manuscript, I would like to propose a few revisions.

1. In this study, authors have predominantly employed the anti-PA monoclonal antibody Raxibacumab for methodological validation. To substantiate the wide applicability of this method, it would be more convincing if an evaluation of another anti-PA monoclonal antibody, Obiltoxaximab (Anthim), along with Anthrax Immunoglobulin Intravenous (AIGIV), could be included.

2. The current anthrax vaccine incorporates PA as a major component, and the anti-PA neutralizing antibodies produced post-immunization play a pivotal role in providing protection. So, whether this method could be extended to assess the immunogenicity of anthrax vaccines. I would encourage authors to consider, discuss, and possibly evaluate this prospect.

3. There are several discrepancies in the presentation and usage of figures in this manuscript that necessitate careful revision and correction. For example, Figure 3D and Figure 4C appear to be duplicated, and the caption for Figure 3 does not accurately label parts A, B, and C.

4. While this manuscript has provided a comprehensive background and cited relevant literatures, updating the references, particularly with the most recent related research, could enhance the value of this work.

The overall writing of the paper is smooth and clearly expresses its meaning.

Author Response

  1. In this study, authors have predominantly employed the anti-PA monoclonal antibody Raxibacumab for methodological validation. To substantiate the wide applicability of this method, it would be more convincing if an evaluation of another anti-PA monoclonal antibody, Obiltoxaximab (Anthim), along with Anthrax Immunoglobulin Intravenous (AIGIV), could be included.

 Response: We agree that it would have been very interesting to utilize several antibodies for comparison purposes. Unfortunately, these clinical reagents are extremely difficult to obtain, as they are stored in the Strategic National Stockpile and are not generally available. The primary goal of this study was to develop and qualify a new cell-based AP-1 reporter assay for anti-PA antibodies, and expired raxibacumab was the only standard available to us for qualification. It is our future goal to compare the potency of raxibacumab, obiltoxaximab and AIGIV when these clinical grade reagents become available to us. Qualification of this assay is one major step in advancing to this objective.

2. The current anthrax vaccine incorporates PA as a major component, and the anti-PA neutralizing antibodies produced post-immunization play a pivotal role in providing protection. So, whether this method could be extended to assess the immunogenicity of anthrax vaccines. I would encourage authors to consider, discuss, and possibly evaluate this prospect.

Response: We note that qualification and validation in this setting would involve new experiments incorporating the matrix of human serum into the design of the assay. We thank you for this suggestion, and we have added discussion regarding the potential use of this method for evaluating the immunogenicity of anthrax vaccines. This is a very logical next step for our research program.

3. There are several discrepancies in the presentation and usage of figures in this manuscript that necessitate careful revision and correction. For example, Figure 3D and Figure 4C appear to be duplicated, and the caption for Figure 3 does not accurately label parts A, B, and C.

Response: We thank you for having discovered this mistake and apologize for having used a wrong version of Figure 3 when the figures were inserted during conversion into the Word file format during the submission process. We revised the Figure 3 and added A, B and C to the caption.

4. While this manuscript has provided a comprehensive background and cited relevant literatures, updating the references, particularly with the most recent related research, could enhance the value of this work.

Response: As requested by you and other referees, more recent references were added to the manuscript.

Reviewer 4 Report

The paper shows the establishment of a new cell based assay to detect the functionality of anti-anthrax Toxin Antibodies. The authors generated a stable reporter cell line to measure c-Jun activity. The transcription factor c-Jun is degraded following inactivation of the MKK pathway.

The paper is clearly written and the data convincing.

The only but clearly missing point of the manuscript is the direct comparison of the new method with existing methods for the same question. Is the reporter method more sensitive, more robust, quicker....?

Author Response

Response: We thank you for your suggestion to directly compare the performance of the new method and existing methods. The following sentences are added to the discussion.

Assay qualification results show that the new AP-1 reporter assay has similar repeatability and intermediate precision when compared to the published cAMP assay and TNA. However, the cAMP assay was validated using a self-developed human-anti-PA monoclonal antibody and the TNA was validated using human anti-anthrax vaccine adsorbed (AVA). Therefore, a side-by-side study using the same testing material would be necessary to definitively compare AP-1 reporter, TNA and cAMP assay performance.  

Round 2

Reviewer 2 Report

The manuscript has been improved.